# Treatment of Non-Alcoholic Steatosis: Preclinical Study of a New Nutraceutical Multitarget Formulation

**DOI:** 10.3390/nu12061819

**Published:** 2020-06-18

**Authors:** Laura Micheli, Alessandra Pacini, Lorenzo Di Cesare Mannelli, Elena Trallori, Roberta D’Ambrosio, Carlo Bianchini, Pietro Lampertico, Carla Ghelardini

**Affiliations:** 1Department of Neuroscience, Psychology, Drug Research and Child Health-Neurofarba—Pharmacology and Toxicology Section, University of Florence, 50139 Florence, Italy; laura.micheli@unifi.it (L.M.); trallorielena@gmail.com (E.T.); carla.ghelardini@unifi.it (C.G.); 2Department of Experimental and Clinical Medicine, Anatomy and Histology Section, University of Florence, 50134 Florence, Italy; alessandra.pacini@unifi.it; 3Foundation IRCCS Ca’ Granda Ospedale Maggiore Policlinico—Division of Gastroenterology and Hepatology—CRC “A. M. and A. Migliavacca” Center for Liver Disease, 20122 Milan, Italy; roberta.dambrosio@policlinico.mi.it (R.D.); pietro.lampertico@unimi.it (P.L.); 4Apharm srl, 28041 Arona, Italy; carlo.bianchini@apharm.it; 5Department of Pathophysiology and Transplantation, University of Milan, 20122 Milan, Italy

**Keywords:** fatty liver, non-alcoholic fatty liver disease (NAFLD), non-alcoholic steatohepatitis (NASH), polyunsaturated fatty acid (PUFA), *Silybum marianum* extract, choline, *Tanacetum parthenium* extract, green coffee Arabic extract, dl-α-tocopheryl acetate, AP-NHm

## Abstract

Multifactorial pathogenesis of non-alcoholic steatohepatitis (NASH) disease, a wide-spread liver pathology associated with metabolic alterations triggered by hepatic steatosis, should be hit by multitarget therapeutics. We tested a multicomponent food supplement mixture (AP-NHm), whose components have anti-dislipidemic, antioxidant and anti-inflammatory effects, on in vitro and in vivo models of NASH. In vitro, hepatic cells cultures were treated for 24 h with 0.5 mM oleic acid (OA): in the co-treatment set cells were co-treated with AP-NH mixtures (AP-NHm, 1:3:10 ratio) and in the post-injury set AP-NHm was added for 48 h after OA damage. In vivo, C57BL/6 mice were fed with high-fat diet (HFD) for 12 weeks, inducing NASH at 7th week, and treated with AP-NHm at two dosages (1:3 ratio) in co-treatment or post-injury protocols, while a control group was fed with a standard diet. In in vitro co-treatment protocol, alterations of redox balance, proinflammatory cytokines release and glucose uptake were restored in a dose-dependent manner, at highest dosages also in post-injury regimen. In both regimens, pathologic dyslipidemias were also ameliorated by AP-NHm. In vivo, high-dose-AP-NHm-co-treated-HFD mice dose-dependently gained less body weight, were protected from dyslipidemia, and showed a lower liver weight. Dose-dependently, AP-NHm treatment lowered hepatic LDL, HDL, triglycerides levels and oxidative damage; co-treatment regimen was anti-inflammatory, reducing TNF-α and IL-8 levels. Hepatic lipidic infiltration significantly decreased in co-treated and post-injury-AP-NHm-HFD animals. The multitarget approach with AP-NHm was effective in preventing and reducing NASH-related pathologic features, warranting for the clinical development of this compound.

## 1. Introduction

Non-alcoholic fatty liver disease (NAFLD) is a chronic liver disease which is not associated with excessive alcohol consumption and whose worldwide prevalence is estimated to be 25%, with regional and sex-related differences [1,2,3]. In one third of cases it may progress to a more severe liver pathology termed non-alcoholic steatohepatitis (NASH), associated with inflammation, cell injury, hepatocyte ballooning, and liver fibrosis [4,5]. Hepatocellular lipid overload results in lipotoxicity, causing a sublethal or lethal injury. Release of pro-inflammatory cytokines starts a feed-forward loop of inflammation which makes it chronic and leads to hepatic tissue remodeling and fibrosis. In this pathological picture, oxidative stress, mitochondrial dysfunction, and insulin resistance are the main players [4]. The public health impact of NASH is major because of the high risk of life-threatening liver-related outcomes, such as cirrhosis and hepatocellular carcinoma (HCC), as well as of major systemic manifestations (i.e., diabetes, cardiovascular damage, etc.). Despite the significant burden of this pathology, no therapy was approved by Federal Drug Administration (FDA) or European Medicines Agency (EMA) and, currently, the cornerstone of NASH management is life-style modifications: eating healthy food and doing regular exercise. Current pharmacological suggestions are based on drugs usually addressed to treatment of type 2 diabetes or dyslipidemia, which may be administered following strictly personalized criteria: for instance, the antioxidant vitamin E and the PPARγ-agonist pioglitazone [6,7,8,9,10]. Several agents targeting specifically different stages and molecular events of this pathology are in clinical trials [11], such as obeticholic acid (OCA), a Farnesoid X Receptor (FXR) agonist [12,13] and elafibranor, a PPAR-α/δ agonist [14]. In the present study, we tested the curative and preventive properties of a multi-nutraceutical formulation (AP-NHm), composed of: *Silybum marianum* extract (80% silymarin), the omega-3 fatty acid docosahexaenoic acid (DHA), choline, *Tanacetum parthenium* (feverfew) extract (0.1% parthenolides), green coffee Arabic extract (45% chlorogenic acid), DL α-tocopheryl acetate (vitamin E). These compounds are or contain nutrients known to intervene on different aspects of the disease: lipid load, inflammatory processes, oxidative unbalance and consequent glucose metabolism alteration. Silymarin is the principal component of the medicinal plant *Silybum marianum*: in a juvenile model of NASH, its supplementation to the diet ameliorated glucose and lipid profiles, reduced fibrosis, inflammation and oxidation [15]. Docosahexaenoic acid (DHA) is an omega-3 fatty acid deriving from marine sources; according with previous evidence, it is low-concentrated in the blood and liver of NASH patients and its supplementation in pre-clinical and clinical studies inhibited the progression of hepatic steatosis and showed hypotriglyceridemic properties, thus ameliorating nutritional NAFLD [16,17]. Choline is an indispensable nutrient whose metabolism is regulated by the liver and whose dietary deficiency paves the way to fatty liver and its more aggressive declinations: European Food Safety Agency (EFSA) defined it as the food ingredient necessary for a “healthy functioning liver” [18,19]. Parthenolide is the main sesquiterpene lactone of the curative plant *Tanacetum parthenium* (feverfew): it is a well-known anti-inflammatory phytoagent with also proved anti-fibrotic activity in liver disease [20,21]. Chlorogenic acid is the major phenolic component of green coffee beans: the extract was showed to have antidiabetes, anti-lipidemic, anti-obesity properties [22]. Vitamin E is a well-known antioxidant, tested for hepatic steatosis in PIVEN trial with satisfying results [7] and, at present, could be given as therapy for NASH to selected categories of patients.

AP-NHm was tested in vitro and in vivo. In vitro experiments were performed on the human hepatocellular carcinoma cell line HepG2, which was treated with oleic acid: on this model of steatosis [23,24,25] we tested the hepatoprotective effect of AP-NHm at three different increasing concentrations (AP-NHm 1, AP-NHm 2, AP-NHm 3 in 1:3:10 ratio), both in co-treatment and post-injury regimens. The nutraceutical mixture was concentration-dependently able to reduce the complex metabolic damage induced by lipid load and, interestingly, the product was active both in preventing alteration and in reducing the established damage. The in vivo studies tested the beneficial effects of AP-NHm at two different increasing concentrations (1:3 ratio) in a mouse model of NASH, induced by prolonged feeding with a high-fat diet (HFD, 60% fat), in two protocols of administrations: preventive (co-treatment) or therapeutic (post-injury). Protection was achieved in a dose-dependent manner: an almost complete prevention was obtained at higher concentrations, when daily administered from day one in co-treatment; a significative recovery was also evidenced in rodents treated immediately after the onset of metabolic alterations.

## 2. Materials and Methods

### 2.1. Cell Culture and Steatosis Model

The steatosis model was reproduced in vitro using the human hepatocellular carcinoma cell line HepG2. Cells were treated with 0.5-mM oleic acid (in 1% BSA; OA/BSA) for 24 h [23,24,25]. The cell line HepG2 (hepatocellular carcinoma cell line) was obtained by American Type Culture Collection (Manassas, VA, USA). Cells were cultured in Dulbecco’s modified Eagle’s medium (DMEM; GIBCO Life Technologies, Grand Island, NY, USA) with 10% (vol/vol) bovine fetal serum in the presence of 1% antibiotic–antimycotic solution at 37 °C with 5% CO2. Confluent cells were treated with 0.5-mM oleic acid in the presence of 1% albumin for 24 h. Control condition was performed allowing incubation with albumin. The effect of 3 differently concentrated AP-NHm co-incubated with OA/BSA (24 h; co-treatment protocol) or incubated for 48 h suddenly after 24 h treatment with OA/BSA (post-injury protocol) was evaluated.

### 2.2. Animals

Male C57BL/6 mice (Envigo, Varese, Italy) weighing approximately 20 g at the beginning of the experimental procedure were used. Animals were used in CeSAL (Centro Stabulazione Animali da Laboratorio, University of Florence) and used at least one week after their arrival. Twelve mice were housed per cage (size 26 × 41 cm) kept at 23.0 ± 1.0 °C with a 12 h light/dark cycle, with lights on at 7 a.m.; during acclimatization they were fed a standard laboratory diet and tap water ad libitum. All animal manipulations were carried out according to the Directive 2010/63/EU of the European parliament and of the European Union council (22 September 2010) on the protection of animals used for scientific purposes. The ethical policy of the University of Florence complies with the Guide for the Care and Use of Laboratory Animals of the US National Institutes of Health (NIH Publication no. 85–23, revised 1996; University of Florence assurance number: A5278-01). Formal approval to conduct the experiments described was obtained from the Animal Subjects Review Board of the University of Florence. Experiments involving animals were reported according to ARRIVE guidelines [26]. All efforts were made to minimize animal suffering and to reduce the number of animals used.

### 2.3. Induction of Metabolic Syndrome

Metabolic syndrome was induced feeding the animals with a high-fat diet (HFD; Research Diets, New Brunswick, NJ, USA) for 12 weeks ad libitum. The model is consistent, with minor modification, to what was previously published [27,28]. HFD contains: 60% fat, 20% protein and 20% carbohydrate, as a percentage of total Kcal [29]. Control animals were fed ad libitum for 12 weeks with a standard diet (24% protein, 58% carbohydrate, 18% fat; Envigo, Varese, Italy).

### 2.4. Preparation of AP-NHm

The nutraceutical formulation AP-NHm based on vegetal extracts contains:pure omega 3 fatty acid (DHA) (Sochim International S.p.A., Cornaredo, Milan, Italy);pure choline bitartrate (Giusto Faravelli S.p.A., Milan, Italy);hydroalcoholic dry extract prepared by means of aqueous extraction and alcoholic precipitation (30:70) from *Tanacetum parthenium*, characterized by 0.1% parthenolides (EPO S.r.l., Milan, Italy);hydroalcoholic dry extract prepared by means of aqueous extraction and alcoholic precipitation (50:50) from *Silybum marianum*, characterized by 80% silymarin (VIVATIS PHARMA Italia S.r.l., Varese, Italy);hydroalcoholic dry extract prepared by means of aqueous extraction and alcoholic precipitation (50:50) from green coffee beans, characterized by 45% chlorogenic acid and 5% caffeine (VIVATIS PHARMA Italia S.r.l., Varese, Italy);DL α-tocopheryl acetate: 7.5 µg/mL.

### 2.5. Treatments

In vitro experiments were carried on a model of steatosis reproduced on the human hepatocellular carcinoma cell line HepG2 treated with 0.5 mM oleic acid (in 1% BSA; OA/BSA) for 24 h [23,24,25]. We evaluated the hepatoprotective effect of AP-NHm at 3 different growing concentrations (AP-NHm 1, AP-NHm 2, AP-NHm 3 in 1:3:10 ratio) in two protocols: co-treatment (co-incubation with OA/BSA for 24 h) and therapeutic or post-injury (48 h of recovery incubation with AP-NHm immediately after 24 h treatment with OA/BSA). AP-NHm 1, 2 and 3 presented the same proportion of components at different concentrations as depicted in Table 1. In vivo, HFD-mice were divided into two arms: co-treatment (treating simultaneously with HFD feeding) and post-injury (treating started with NASH onset). In both cases, mice were treated with AP-NHm 1 or 2, low dose and high dose, respectively, differing in concentration (1:3 ratio) as shown in Table 1. Mixtures were suspended in 1% carboxymethylcellulose sodium salt and *per os* administered. The administration was divided in two parts: a morning treatment (1 h after the light phase of the circadian light/dark cycle in the animal facility) and an evening treatment (1 h before the dark phase), as explained by Appendix A.

### 2.6. Cell Viability (MTT)

Cell viability was evaluated by the reduction of 3-(4,5-dimethylthiozol-2-yl)-2,5-diphenyltetrazolium bromide (MTT) as an index of mitochondrial compartment functionality. Post treatments, after three extensive washing, 1 mg/mL MTT was added into each well and incubated for 2 h at 37 °C. After washing, the formazan crystals were dissolved in 100 μL dimethyl sulfoxide. The absorbance was measured at 580 nm. Experiments were performed in quadruplicate on at least 3 different cell batches.

### 2.7. Measurements of Superoxide Dismutase (SOD) Activity

Total SOD activity was determined by monitoring the inhibition of the reduction of ferricytochrome C at 550 nm, using the xanthine–xanthine oxidase system as the source of superoxide. One unit of the SOD is defined as the amount of the enzyme required to inhibit 50% of the rate of cytochrome-C reduction [30]. Catalase activity was measured by following the rate of H2O2 consumption spectrophotometrically at 240 nm.

### 2.8. Evaluation of Lipid Peroxidation in Cellular Model (Thiobarbituric Acid Reactive Substances, TBARS)

After treatments, cells were scraped, and cell suspension underwent a freeze/thaw cycle. The suspension was added to 4 mL reaction mixture consisting of 36-mM thiobarbituric acid (Sigma-Aldrich, Milano, Italy) solubilized in 10% CH3COOH, 0.2% SDS, pH was adjusted to 4.0 with NaOH. The mixture was heated for 60 min at 100 °C and the reaction was stopped by placing the vials in ice bath for 10 min. After centrifugation (at 1600× g at 4 °C for 10 min) the absorbance of the supernatant was measured at 532 nm (PerkinElmer spectrometer, PerkinElmer, Waltham, MA, USA) and TBARS were quantified in µmoles/milligram of total protein using 1,1,3,3-tetramethoxypropane as standard.

### 2.9. Evaluation of Proinflammatory Cytokines

TNF-alfa and IL-8 were measured in the culture medium and in plasma by using a commercially available enzyme-linked immunosorbent assay (BioSource, Bergamo, Italy).

### 2.10. Evaluation of Glucose Uptake

The cell uptake of 6-hydroxy glucose was measured via colorimetric reaction (Abcam, Milano, Italy).

### 2.11. Evaluation of Steatosis Grade by Oil Red Staining

After treatments cells were fixed in 4% formaldehyde for 15 min, then washed in PBS and incubated for 5 min with 60% isopropanol. Cells were treated with 0.1% oil red solution for 1 h. After an additional wash images were captured under a light microscope and optical density (O.D.) was measured.

### 2.12. Evaluation of Total Cholesterol and Triglycerides

After treatment, lipids were extracted from cells with chloroform: isopropanol: tween-20 solution (7:11:0.1). Enzymatic methods applied to commercial kit were used to perform specific quantitative measurements (BioSource, Italy).

### 2.13. Body Weight and Food Intake

During the experiment, the body weight of mice was measured every 3 days while the food intake was measured weekly.

### 2.14. Collection of Blood and Analytical Methods

Blood was collected from the facial vein of the mice under light ether anesthesia into Eppendorf tubes containing heparin (20 µL, 25,000 IU/5 mL) weekly starting from week 1. Samples were analyzed for glucose, triglycerides and total cholesterol levels using a Reflotron reflectance photometric analyzer (Reflotron, Roche Diagnostics, Monza, Italy). LDL and HDL cholesterol were measured using the same method only on week 12 (Day 84).

### 2.15. Glucose Tolerance Test

On week 12, glucose tolerance test was performed by intraperitoneally injections of glucose (1 mg kg^−1^). Blood glucose was measured by Accu-Chek Aviva (Roche Diagnostics, Monza, Italy) at 0 min, 15 min and 30 min after glucose injections.

### 2.16. Tissues Explant

On week 12 (Day 84), after the behavioral measurements and the biochemical analysis, mice were sacrificed by cervical dislocation and tissues explant was performed. In particular, perigonadal fat pad, liver, heart and kidney were collected and weighted.

### 2.17. Liver Histopathology

Liver tissue samples were fixed in 4% formalin and processed for embedding and sectioning. Liver sections (5 µm) were stained with hematoxylin and eosin (H&E). The quantitative scoring of H&E-stain-based liver tissue sections was conducted according to the previous methods with slight modifications, regarding steatosis [31,32]. Briefly, each section was examined by a specialist who was blinded to the sample information and hepatic steatosis and inflammation scores were evaluated. The degree of steatosis was graded ‘0’ to ‘10’ based on the average percent of fat-accumulated hepatocyte per field at 4 or 20 × magnification under H&E staining (grading 0, <5% of fat accumulation and grading 10,  >75% of fat accumulation).

### 2.18. Lipid Levels in Liver

Liver homogenate was dosed for triglycerides, LDL and HDL cholesterol using a Reflotron reflectance photometric analyzer (Reflotron, Roche Diagnostics, Monza, Italy).

### 2.19. Adipose Tissue Histopathology

The perigonadal fat pad was collected for histological and biochemical analyses. For histology, the tissues were drop-fixed in 10% neutral buffered formalin for 2–3 days, after which they were processed for paraffin embedding. Five-micrometer sections were then cut, collected, stained with hematoxylin and eosin and digitized. Adipocyte size was measured by an investigator blinded to the experimental grouping in × 40 microscope fields by counting the total number of adipocytes within predefined area grids and then dividing the area by the total number of adipocytes within the grids to calculate average adipocyte size. For each sample, three replicate tissue sections were analyzed, with three fields counted in each section, for a total of nine fields averaged per sample.

### 2.20. Evaluation of Lipid Peroxidation in Tissue (Thiobarbituric Acid Reactive Substances, TBARS)

Thiobarbituric acid-reactive substances (TBARS) assay was assessed as an index of lipid peroxidation according to [33]. The TBARS determination was carried out in mice liver and heart homogenate obtained from tissue homogenized in PBS at the final concentration of 10% *w/v*. Then were added FeCl_3_ (20 μM, Sigma-Aldrich, St. Louis, MO, USA) and ascorbic acid (100 μM, Sigma-Aldrich) to obtain the Fenton reaction. At the end of incubation, the mixture was added to 4 mL reaction mixture consisting of 36-mM thiobarbituric acid (Sigma-Aldrich) solubilized in 10% CH3COOH, 0.2% SDS, and pH was adjusted to 4.0 with NaOH. The mixture was heated for 60 min at 100 °C and the reaction was stopped by placing the vials in an ice bath for 10 min. After centrifugation (at 1600 g at 4 °C for 10 min) the absorbance of the supernatant was measured at 532 nm (PerkinElmer spectrometer) at 550 nm (PerkinElmer spectrometer) and TBARS were quantified in µmol/mg of total proteins using 1,1,3,3-tetramethoxypropane as the standard.

### 2.21. Carbonylated Proteins

Carbonylated proteins were evaluated in liver and heart tissue homogenate. Tissue proteins extract was quantified by BCA. Twenty micrograms of each sample were denatured by 6% SDS and derivatized by 15-min incubation with 2–4 dinitrophenyl hydrazine (DNPH; Sigma-Aldrich, Italy) at room temperature. Samples were separated on a 4–12% sodium dodecyl sulfate (SDS)-polyacrylamide gel by electrophoresis and transferred onto nitrocellulose membranes (BioRad, Italy). Membranes were blocked with 5% nonfat dry milk in phosphate-buffered saline (PBS) containing 0.1% Tween-20 (PBST) and then probed overnight with primary antibody specific versus DNPH (Sigma-Aldrich, Italy) 1:5000 in PBST/5% nonfat dry milk. After washing with PBST, the membranes were incubated for 1 h in PBST containing the appropriate horseradish peroxidase-conjugated secondary antibody (1:5000; Cell Signaling, Danvers, MA, USA) and again washed. ECL (Pierce, USA) was used to visualize the peroxidase-coated bands. Densitometric analysis was performed using the “ImageJ 1.42” image analysis software (NIH, Bethesda, ML, USA). For each experiment, the density of all bands showed in a lane was reported as mean. The total protein amount quantified by Ponceau staining was used to normalize all samples.

### 2.22. Statistical Analysis

Each in vitro experiment was performed at least three times. In vivo measurements were performed on 12 animals for each group, analyzed in 2 different experimental arms (co-treatment and post-injury) by researchers blinded to the treatment procedure. Results were expressed as means ± SEM and the analysis of variance was performed by ANOVA test. A Bonferroni’s significant difference procedure was used as a *post hoc* comparison. *p-*values less than 0.05 were considered significant. Data were analyzed using the ‘‘Origin 8.1” software (Northampton, MA, USA).

## 3. Results

### 3.1. In Vitro Experiments

#### 3.1.1. AP-NHm Protects Cellular Proliferative Abilities

Human hepatocellular carcinoma cell viability and proliferation were tested in both co-treatment and post-injury protocols. Incubation of HepG2 cells for 24 h with OA/BSA did not significantly modify cell viability (Figure 1a), while it negatively influenced cellular proliferative activity, since it decreased by one third compared to control cells. In post-injury protocol, treatment with AP-NHm 2 and 3 for 48 h almost completely restored the proliferative capabilities (Figure 1b).

#### 3.1.2. AP-NHm Restores Mitochondrial SOD Activity and Ameliorated Lipid Peroxidation Status

OA/BSA treatment altered oxidative stress defense system and redox balance, as showed by mitochondrial superoxide dismutase (SOD) activity and lipid peroxidation status (Figure 2a–d). Lipid overloading reduced antioxidant properties of mitochondria since activity of SOD was almost halved after 24 h of incubation. In the preventive co-treatment protocol, harm was significantly attenuated by AP-NHm 1 and even totally canceled by AP-NHm 2 and 3 (Figure 2a); nevertheless, only AP-NHm 2 and 3 were able to re-activate mitochondrial enzyme at the end of the 48 h treatment following the insult (Figure 2b). Treatment with OA/BSA induced an increase of TBARS (thiobarbituric acid reactive substances) 24 h after treatment and after 48 h of recovery. One-day co-incubation with the three AP-NHm doses induced a significant decrease in TBARS in a dose-dependent manner (Figure 2c). The repairing effect was instead less intense after the two-day incubation with AP-NHm 3 and totally absent in the treatment with AP-NHm 1 and 2 (Figure 2d).

#### 3.1.3. AP-NHm Obstacles Inflammatory Events

Inflammatory events were assessed by measuring TNF-α and interleukin-8 (IL-8) in the culture medium. A high inflammation state caused by lipid overloading for 24 h was showed by elevated cytokines levels in both protocols, even if a weak decrease of inflammation occurred 2 days after the end of the insulting treatment. Preventive co-incubation and post-injury recovery incubation of 48 h with the supplementary mixture AP-NHm reduced TNF-α and IL-8 levels in a concentration-dependent manner (Table 2). The highest concentration of components (AP-NHm 3) almost halved the pathologic levels of cytokines in both regimens.

#### 3.1.4. AP-NHm has Protective Effects on Metabolic Alteration

We investigated the protective abilities of the three doses of AP-NHm in the most relevant signatures of metabolic syndrome: alteration of glucides, triglycerides and cholesterol. Carbohydrate metabolism was tested through cell glucose uptake after insulin stimulation of 1-µM (Table 3). Following OA/BSA incubation, insulin sensitivity was significantly reduced (40% less after 1 day). In the post-injury treatment insulin sensitivity was 70% less at the end of the recovery time, free from lipid solution. The administration of AP-NHm exhibited a concentration-dependent rescuing effect, reaching an almost complete recovery with the third composition.

#### 3.1.5. AP-NHm Reduces and Impairs Intracellular Lipid Deposits

Intracellular triglycerides and cholesterol deposits were enlarged by treatment with the combination of OA/BSA, an effect that was observed by analyzing cellular triglycerides/protein (Figure 3a,b) and cholesterol/protein (Figure 3c,d) ratio. Following incubation with OA/BSA, triglycerides/protein ratio was about four times higher than the control group. Co-incubation (Figure 3a) and post-injury-incubation (Figure 3b) with AP-NHm produced different effects: during the co-incubation only AP-NHm 3 was able to reduce lipid presence; in the post-injury treatment of 48 h AP-NHm 2 and 3 reduced the lipid deposits in a concentration-dependent manner, since the latter almost halved the ratio. Lipid overload increased the concentration of total cholesterol (Figure 3c,d), but in both treatment protocols a protective effect of AP-NHm 2 and 3 was found. Indeed, the co-incubation with AP-NHm 2 and 3 lowered the cholesterol levels to normal values (Figure 3c) in preventive experiments. In post-treatment (Figure 3d), at the end of the 48-hour recovery, the cholesterol levels decreased in a concentration-dependent manner in all three treatment groups.

#### 3.1.6. AP-NHm Improves Steatosis Grade

Oil Red staining of cells highlighted lipid accumulation, pointing to the respective steatosis grade shown as optical density (O.D.) analysis (Figure 4a,b). OA/BSA treatment strongly increased the steatosis grade of cells, but the damage was widely reduced by co-treatment with AP-NHm 3 (Figure 4a) and by post-injury incubation with AP-NHm 2 and 3 (Figure 4b).

### 3.2. In Vivo Experiments

#### 3.2.1. AP-NHm 2 Co-Treatment Results in Weight Loss

Animals were fed for 12 weeks ad libitum with either standard diet or high-fat diet and first signs of metabolic disorders appeared after 6–7 weeks of HFD: in those mice, on Day 42 the blood levels of total cholesterol, triglycerides and glucose were significatively increased with respect to the standard diet fed animals (Appendix A). Following the protocols used for in vitro experiments, two protocols were scheduled also for in vivo treatments: a co-treatment with two different concentrations of AP-NHm, administered simultaneously with HFD feeding from week 1 to week 12 (Day 1 to Day 84), and a post-injury treatment, beginning on week 7 (on Day 49), once hepatosteatosis was established, as confirmd by metabolic alterations. Control groups of both diets (standard and fat-rich) were treated with vehicle (1% carboxymethyl cellulose sodium salt). Body weight of rodents from each experimental group were monitored for the duration of the study (Figure 5a): HFD-fed animals showed a significant weight increase starting from Day 30; the HFD group co-treated with AP-NHm 2 exhibited a loss of body weight from Day 41; the same concentration of nutrients induced a remarkable reduction of body weight from Day 66 to 78 in the post-injury protocol. The AP-NHm 1, in both protocols, was ineffective on this parameter. The evaluation of food intake showed that high-fat diet determined its reduction with no influence from the supplementation with AP NHm (Figure 5b).

#### 3.2.2. AP-NHm has Hypocholesterolemic, Hypolipidemic and Hypoglycemic Effects

Blood levels of total cholesterol, triglycerides and glucose were measured every week to monitor the typical metabolic alterations of this disease model that, starting from Day 42, showed a significant increase in comparison to normal diet-fed group, thus leading to set the beginning of the post-injury treatment on Day 49, i.e., when hepatosteatosis syndrome was firmly settled. In Table 4 the effects of both AP-NHm mixtures in the co-treatment and post-injury protocols (in table marked as groups 3–4 and 5–6, respectively) were reported till Day 84. In co-treatment AP-NHm 2 was able to fully normalize total cholesterol over the 6 weeks and to restore LDL levels on Day 84, while HDL was higher than control levels; AP-NHm 1 had an hypocholesterolemic activity on days 70 and 77. Post-injury regimen was effective with only AP-NHm 1 starting from Day 70 until Day 84 and on the last day also LDL appeared ameliorated, whereas HDL was unchanged. A similar trend was observed for triglyceride blood levels whose oscillations were due to the different treatment modalities and dose of mixture supplemented to the animals. Co-treatment with both AP–NH mixtures was effective from Day 49 till the end of the experiment, but more striking results emerged in the group treated with high-dose AP-NHm; post-injury administrations significantly reduced lipid levels from Day 70 with AP-NHm 2, from Day 77 with AP-NHm 1. Hyperglycemia was not totally abolished by supplementations, but it was diminished: simultaneous administration of AP-NHm 2 with fat-rich diet caused an about 20% reduction from Day 49 till the end of the study, while in post-injury protocol the same AP-NHm 2 was effective from Day 70; AP-NHm 1 did not induce any amelioration. Similar results were observed in metabolism response to glucose load, via oral glucose tolerance test: HFD feeding altered metabolism ability to catch glucose, since an increasing hyperglycemia was set following glucose administration. As in hyperglycemia test, supplementation with AP-NHm 1 was ineffective, while co-treatment with AP-NHm 2 showed beneficial results also in post-injury protocol, even if with lesser extent results.

#### 3.2.3. AP-NHm has Anti-Inflammatory Effects

At the end of the study, blood samples were collected from all groups and two cytokines were titrated: TNF-α and IL-8 (Table 5). Both molecules resulted more concentrated in blood of HFD-fed rodents than control hematic samples. Co-treatment protocol with both AP-NHm mixtures was able to prevent an excessive cytokine concentration increase, even though the high-dose AP-NHm worked better; post-injury treatment produced the same effect when dosed as AP-NHm 2.

#### 3.2.4. AP-NHm Reduces Abnormal Heart, Liver and Adipose Tissue Weight

All the rodents were sacrificed and ex-vivo analysis was performed. The weight of the liver, heart, kidney, and perigonadal fat pad, organs that underwent to the fat-rich diet-dependent metabolic disorders were recorded (Figure 6a–d). Kidney weight was not affected neither by HFD feeding, nor by supplementations, while significant changes in the weight of the other organs were evidenced. Liver weight was dramatically increased (about 1000 mg) by HFD feeding, and the two supplementation formulas prevented this hepatic fattening. In both regimens AP-NHm 2 was the most effective, because its administration from the study onset completely prevent weight gain and the therapeutic treating cut down liver mass increase; however, AP-NHm 1 was poor active and only when administrated concomitantly to the fat-rich diet. The mass of the heart grew of 50% and only co-treatment with high-dose AP-NHm 2 was able to counteract this pathologic effect. Under fat-rich diet, also perigonadal fat weight, doubled (normalized to body weight) and this alteration was attenuated only by administrating AP-NHm 2 in co-treatment regimen.

#### 3.2.5. High dose AP-NHm Significantly Resizes Adipocyte Area of Perigonadal Fat Pad

Subsequent histological analysis focused on perigonadal adipose tissue and liver. Histological sections of perigonadal adipose tissue (Figure 7a–f) and adipocyte area quantification (Figure 7g) showed a HFD-dependent enlargement of adipocytes. This effect was significantly counteracted by AP-NHm 2 in both treatments.

#### 3.2.6. AP-NHm Protects from Hepatic Fat-Accumulation

Under hematoxylin and eosin staining (Figure 8a–f) the hepatic tissue appeared strongly steatotic in the fat-rich diet group; indeed, in a steatosis index ranging from 0 to 10 scale, where 0 states for <5% and “10” for >75% of fat-accumulated hepatocytes, the hepatic tissue scores “8” steatosis grade per field (Figure 8g). AP-NHm 2 was protective from fat-accumulation at both regimens, although more effective in co-administration, whereas AP-NHm 1 was effective only in co-treatment with the diet.

HFD-diet affects the lipid concentration in hepatic parenchyma homogenate by increasing the LDL and HDL concentration by 5 times and the triglyceride one by 6 times (Table 6). The AP-NHm 2 co-treatment and post-injury protocol were effective on all parameters, whereas the co-treatment AP-NHm 1 protocol was effective only on HDL and triglycerides.

#### 3.2.7. AP-NHm Exerts an Antioxidant Effect on Heart and Liver

Oxidative damage was verified in liver and heart assessing lipid peroxidation and protein carbonylation. In the heart HFD-dependent lipid peroxidation was buffered in post-treatment protocol by both supplementations with AP-NHm (Figure 9a); carbonylation of protein increased of 40%, but it was almost totally restored by both AP-NHm therapeutic administrations (Figure 9b,c). In the liver the fat-rich diet led to significant lipid oxidation (Figure 9d). This oxidation was considerably diminished by AP-NHm 2 in co-treatment with the feeding; whereas the percentage of carbonylated proteins were reduced in co-treatment regimen by mix 1 and strongly reduced by mix 2. In post-injury treatment only AP-NHm 2 exhibited a slight effect (Figure 9e,f).

## 4. Discussion

In the present framework of research, we investigated the therapeutic and preventive effects of a new nutraceutical multicomponent formulation (AP-NHm) on NASH in in vitro and in vivo models. NASH is a liver pathology, the negative and more aggressive evolution of NAFLD, originally caused by lipid accumulation in the hepatocytes [4]. The diffusion of NASH is dramatically increasing and in the next years it is claimed to be the first cause of HCC and liver transplantation, therefore it represents a worldwide health issue [2]. Nowadays, there is no pharmacological management accepted by FDA and EMA and the first line of treatment is nutritional intervention, by changing lifestyle of patients. According to EASL [34], AASL [35] and KASL [36], current medications could be vitamin E and the PPAR–agonist pioglitazone, an insulin sensitizer in use for type 2 diabetes treatment, but only in selected categories of patients due to limiting side-effects, related to higher cancer risks, increased body weight and not univocal results [6,7,8,9,10]. Many other promising drugs are in the pipeline or under clinical trials [11,37], such as the FXR agonist OCA, which improved hepatic histology and induced fibrosis remission of NASH patients in both phase 2 and phase 3 randomized controlled trials with the adverse effect of a worsened cholesterol profile [12,13] and PPAR-α/δ agonist elafibranor, which showed an ameliorating trend of pathologic parameters in subgroups of severe NASH patients [14]. In the field of nutraceutical research many efforts are being made to delineate curative abilities of food bioactive molecules and some nutrients were proved to possess ameliorative properties in metabolic disorders. These are the qualities of those nutraceutical compounds present in AP-NHm as part of extracts or as single molecules: silymarin, docosahexaenoic acid (DHA), choline, parthenolide, chlorogenic acid, dl-α-tocopheryl acetate (vitamin E).

In the pathogenesis of NASH, mitochondrial dysfunction is a consequence of lipotoxicity and in turn causes bad functioning of defensive mechanisms against oxidative stress [4]. Lipid load of hepatocytes reduced superoxide dismutase (SOD) activity and increased lipid peroxidation levels, but co-treatment with AP-NHm restored SOD activity and lowered TBARS levels in a concentration-dependent manner, while the treatment after insult was able to ameliorate the oxidative stress state only when dosed at the highest concentration. In mice, HFD feeding altered redox balance, as showed by elevated levels of lipid peroxidation and carbonylated proteins in liver and heart. High dose AP-NHm 2 was effective at improving oxidative unbalance both in preventive co-treatment and in post-injury protocol, while the less concentrated AP-NHm 1 was able to counteract oxidation mechanism only in co-treatment with HFD feeding. Several works supported the antioxidant activity of the bioactive nutrients contained in AP-NHm. Stellavato et al. treated hepatic cells with a less rich formulation and claimed the good antioxidant property of tocopherol acetate, DHA and silymarin [38]; tocopherol acetate is a well-known antioxidant [39], whose hepatoprotective effects proved in PIVEN clinical trial were achieved at 800 IU/day, a much higher concentrations than dosage in AP-NHm mixture [7]. *Tanacetum parthenium* extract probably did not participate in balancing the altered oxidative equation, because many studies reported the ability of parthenolides to induce apoptosis via oxidative stress, depleting GSH levels and rising intracellular ROS levels [21,40]. Silymarin and chlorogenic acid contributed to enhance antioxidant competence of the cells, as verified also in other cases [41,42].

Lipotoxicity is also at the basis of inflammatory context in which hepatosteatosis established: it sublethally injures hepatic cells and, as a result, proinflammatory cytokines are released. They recruit hepatic immune cells and macrophages which infiltrate hepatic parenchyma and, in turn, exacerbate the inflammatory response, turning it from acute to chronic. In our study, the extremely high levels of the two inflammatory cytokines TNF- and IL-8 in cellular medium and in the blood of rodents revealed the activation of the inflammatory cascade. Their presence was reduced in both experimental sets: in the cellular model, AP-NHm had a concentration-dependent anti-inflammatory effect; in rodents, the high-dose AP-NHm 2 significantly reduced inflammatory cytokines release, both in co-treatment and post-injury treatment. Most of the compounds present in AP-NHm have anti-inflammatory properties, as extensively demonstrated by previous works on anti-inflammatory related disease, not only on NASH or NAFLD [20,42,43,44,45,46].

Establishment of NASH in animals was reached by feeding with fat-rich diet, while in the cells the syndrome was simulated by a lipid-overloading damage lasting 24 h. In both cases, many metabolic alterations were induced and treatment with the multi-nutraceutical AP-NHm was able to reduce them with good results. Insulin resistance is a clinical feature of metabolic syndrome and is part of the pathologic picture of steatohepatitis as well [35,47]. It may manifest as reduced glucose uptake from cells stimulated with insulin, since receptors do not respond appropriately to the hormone or it may appear as persistent hyperglycemia following glucose load, because insulin is not able to exert its hypoglycemic activity. In the in vitro model, co-treatments with all AP-NHm protected from insulin resistance, while the same protective effect was reached during the post-injury recovery period only if incubating cells with medium and high concentrated AP-NHm. Insulin activity was less impaired if animals were treated with AP-NHm 2, both in preventive and in post-injury protocols. Yao et al. demonstrated that silymarin ameliorated insulin resistance in NASH rats and proposed that this natural compound was able to modulate the expression of genes involved in visceral obesity, lipolysis and gluconeogenesis [48]. The same role was not delineated for vitamin E since the biomolecule alone seemed not able to repair damaged insulin response mechanisms [49]. Some mechanisms through which DHA exerts this property emerged from studies on animals: anti-inflammatory effects and modulation of the expression of related gene were proposed [50]; similarly, curbed insulin resistance was obtained by supplementation with only CGA in animals with HFD-induced hepatic steatosis [51].

In the pathologic picture of non-alcoholic steatohepatitis, there are metabolic disorders going along with it: hypertriglyceridemia, hypercholesterolemia and hyperglycemia. In the present study, we proved that AP-NHm was able to protect from these metabolic alterations in both protocols: in the cellular model intracellular triglycerides and cholesterol were reduced by co-treatment and post-injury incubation in a concentration-dependent way; in mice the best results was obtained in co-treatment with AP-NHm 2, while post-injury administrations were protective from 21–27 days after the establishment of the pathology. Most of the components contained in the extracts of AP-NHm were reported to have hypotriglyceremic, hypocholesterolemic and hypoglycemic properties and they were often related to metabolic syndrome management. Cicero and co-authors [52] claimed lipid-lowering effects of some nutraceuticals, pointing out also possible mechanisms of action. For example, silymarin was reported to decrease lipid peroxidation of LDL, by acting as a radical scavenger, to inhibit HMG–CoA reductase in vitro, to reduce hepatic total cholesterol and allow LDL uptake in vivo [52,53]. Vitamin E with its anti-obesity, hypoglycemic and hypocholesterolemic properties was extensively reviewed for use in metabolic syndrome management [54] and was supposed to act as a PPAR-receptor activator, inhibitor of the enzyme HMG–CoA reductase and as a radical scavenger [52]. DHA is another nutraceutical with lipid-lowering properties: Molinar-Toribio and his collaborators reported that combination of EPA and DHA lowered LDL-cholesterol levels in a model of mouse metabolic syndrome [55] and, on the basis of many supporting data, EFSA and American Heart Association sustained that DHA can maintain normal blood TG levels or even reduce them by 25–30% [19,56]. DHA seemed to reduce synthesis of VLDL and the synthesis of new TG, acting on substrate and enzymes. Preclinical and clinical works confirmed the antidiabetic, anti-lipidemic, antioxidant, anti-inflammatory properties of chlorogenic acid [57]; parthenolide was also proved to have anti-inflammatory and hypotriglyceremic effects on the liver of rats [58]. The lipid-lowering properties of these functional foods present in AP-NHm, described so far, are probably responsible for the loss of weight registered in the group of rodents under AP-NHm 2 treatment, both in co-treatment and in post-injury therapeutic regimens. This supposed mechanism of weight loss was supported by the absence of changes in food intake of the slimmed down group, which was unvaried in all the others. The same weight loss was not observed for all the explanted organs. AP-NHm 1 and 2 were able to reduce weight of fatty liver, an almost total rescue was reported when mice were co-treated with the high-dose AP-NHm; heart and perigonadal fat pad masses were downsized only by AP-NHm 2 in the preventive regimen. In parallel, the area of adipocytes of this peculiar fat tissue was reduced in co-treatment with AP-NHm 2 and, in addition, when the same mix was administered following the onset of NASH.

The final consequence of a severely altered lipid profile (triglycerides, total cholesterol, LDL and HDL) and insulin resistance is hepatic steatosis, the precursor and pivotal marker of steatohepatitis. As shown so far, metabolic alterations are so crucial in the pathologic picture of NASH, that Araujo and her collaborators recently proposed a new positive definition of the pathology as metabolism-associated steatohepatitis (MASH) [1], rather than a non-alcoholic disease. The above-mentioned insulin resistance is also responsible for the impairment of lipid metabolism: it usually inhibits the activity of hormone sensitive lipase (HSL) enzyme, but if the enzyme is deranged, triglyceride lipogenesis increases and so do free fatty acids, thus reaching the liver where they accumulate [59]. In our study, hepatic steatosis was histologically quantified by a steatosis index depending on the average percentage of fat-accumulated hepatocytes in the parenchyma. Consistently with the lipid-lowering properties elucidated so far, our nutraceutical mixture AP-NHm was able to reduce the steatosis index in a significant manner with both dosages in the co-treatment regimen and with the highest dosage in the post-injury treatment; moreover, a similar pattern of protection was observed in vitro where fat accumulation in hepatocytes was highlighted by O.D. analysis of Oil Red staining. Many studies confirmed the peculiar ability of the main nutrients of our extracts, that are mixed with single bioactive compounds, to reduce hepatic steatosis: silymarin was proved to attenuate insulin resistance and hepatic steatosis after 3 months of treatment [60]; PIVENS trials demonstrated the therapeutic activity of tocopherol, however limited in its administration due to safety concerns [7]; DHA was reported to be protective in combination with the other polyunsaturated fatty acid EPA [16]; choline-deficiency and its metabolism by gut microbiota were associated with hepatic steatosis development [61]; chlorogenic acid injections reduced lipid accumulation in the liver of HFD fed mice [51]; parthenolide was not directly reported to lower hepatic steatosis, but it was documented to reduce liver triacylglycerol levels [58].

## 5. Conclusions

In the present study, a multicomponent mixture (AP-NHm) was planned for intervening on the different features of non-alcoholic steatohepatitis. AP-NHm contained food bioactive molecules able to prevent lipid load, reduce inflammatory processes, oxidative unbalance, the consequent glucose metabolism alteration and hepatic steatosis. The nutraceutical formulation was tested in a model of steatosis reproduced in vitro using the human hepatocellular carcinoma cell line HepG2 treated with 0.5-mM oleic acid. AP-NHm was able to reduce the complex metabolic damages independent of concentration, being active both in preventing alteration and in reducing damage. The nutraceutical compounds also attenuated several metabolic and histological alterations induced in mice by HFD nutrition. A prolonged feeding with a high fat provision evoked a steatohepatitis state characterized by body weight increase, alteration of lipids and glucose levels, histopathologic damage of liver and adipose tissue as well as a general oxidative and inflammatory condition. The tested formulation induced hepatoprotective effects depending on the dosage: an almost complete prevention was obtained when the high-dose AP-NHm 2 was daily administered from Day 1, in co-treatment regimen. Nevertheless, it was noticed that a significative recovery was also obtained when treatments started after the establishment of the metabolic alterations. Further studies are required to understand timing, dosing and safety of this food extracts formulation, but our results set in the promising way of nutraceutical therapy for NASH management.

## Figures and Tables

**Figure 1 nutrients-12-01819-f001:**
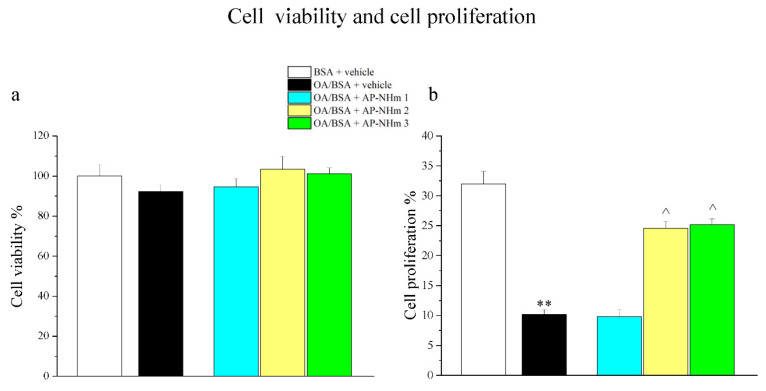
Effects of oleic acid (OA) and AP-NHm on HepG2: cell viability (**a**) and cellular proliferation (**b**). HepG2 were incubated with 0.5-mM oleic acid (OA) in 1% bovine serum albumin (BSA) (OA/BSA) or BSA alone for 24 h in co-treatment or in post-injury recovery treatment (48 h) with AP-NHm 1, 2, 3. ** *p*< 0.01 vs. BSA + vehicle; ^ *p* < 0.05 vs. OA/BSA + vehicle.

**Figure 2 nutrients-12-01819-f002:**
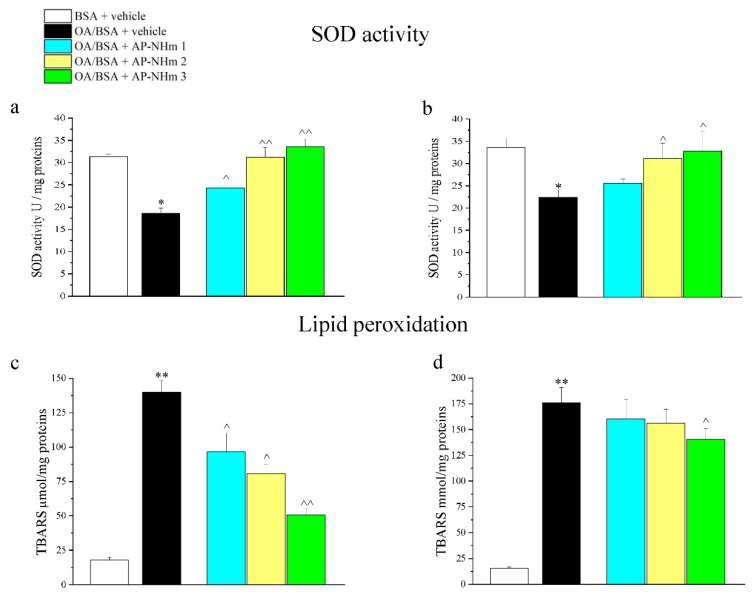
Effects of AP-NHm on oxidative damage: superoxide dismutase (SOD) activity in co-treatment (**a**) and post-injury (**b**) regimens; lipid peroxidation in co-treatment (**c**) and post-injury (**d**) regimens. HepG2 were incubated with 0.5-mM oleic acid in 1% BSA(OA/BSA) or BSA alone for 24 h in co-treatment or in post-injury recovery treatment (48 h) with AP-NHm 1, 2, 3. * *p* < 0.05 and ** *p* < 0.01 vs. BSA + vehicle, ^ *p* < 0.05 and ^^ *p* < 0.01 vs. OA/BSA + vehicle.

**Figure 3 nutrients-12-01819-f003:**
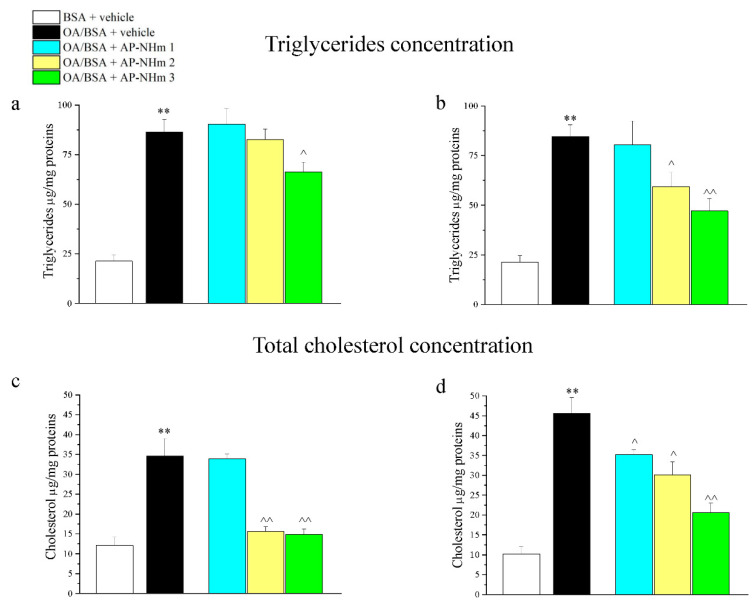
Effects of AP-NHm on dyslipidemias: intracellular triglyceride concentration in co-treatment (**a**) and post-injury (**b**) regimens; total cholesterol concentration in co-treatment (**c**) and post-injury (**d**) regimens. HepG2 were incubated with 0.5-mM oleic acid in 1% BSA (OA/BSA) or BSA alone for 24 h in co-treatment or in post-injury recovery treatment (48 h) with AP-NHm 1, 2, 3. ** *p* < 0.01 vs. BSA + vehicle, ^ *p* < 0.05 and ^^ *p* < 0.01 vs. OA/BSA + vehicle.

**Figure 4 nutrients-12-01819-f004:**
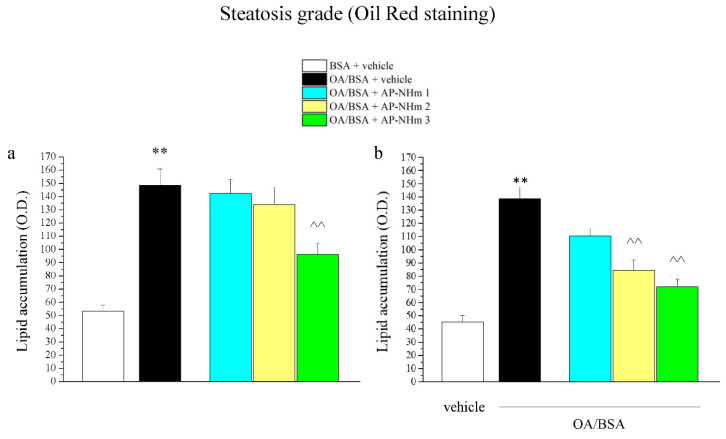
Effects of AP-NHm 3 on steatosis grade measured by oil red staining. HepG2 were incubated with 0.5-mM oleic acid in 1% BSA (OA/BSA) or BSA alone for 24 h in co-treatment or in post-injury recovery treatment (48 h) with AP-NHm 1, 2, 3. (**a**) optical density (OD) of lipid accumulation with AP-NHm in co-treatment regimen; (**b**) OD of lipid accumulation with AP-NHm in post-injury regimen. ** *p* < 0.01 vs. BSA + vehicle, ^^ *p* < 0.01 vs. OA/BSA + vehicle.

**Figure 5 nutrients-12-01819-f005:**
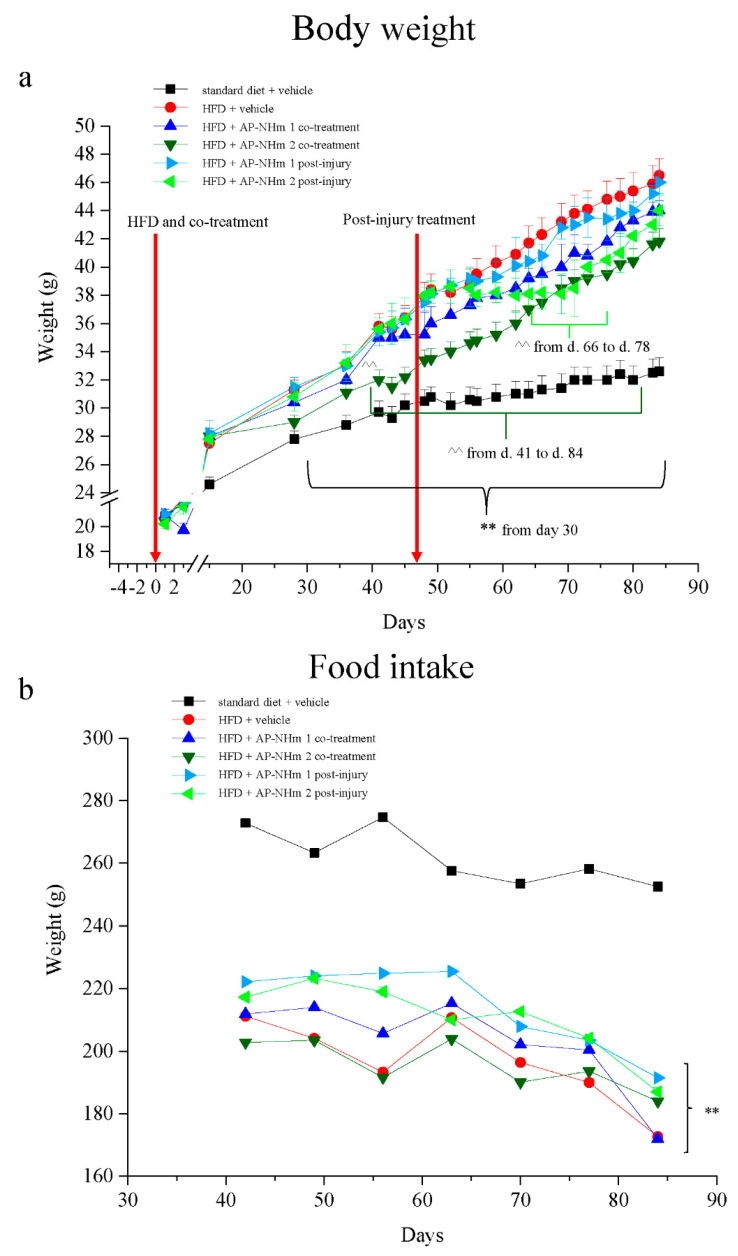
Effects of high-fat diet (HFD) and AP-NHm 1–2, in co-treatment or in post-injury treatment, (**a**) on body weight from Day 0 and (**b**) on food intake from Day 42 to Day 84. Mice were fed with normal or HFD diet from Day 0 to Day 84: two groups were co-treated with AP-NHm while other two groups started AP-NHm treatment on Day 49. Control groups were treated with vehicle throughout the study. Each value represents the mean ± SEM of 12 mice per group. ** *p* < 0.01 vs. standard diet + vehicle, ^^ *p* < 0.01 vs. HFD + vehicle.

**Figure 6 nutrients-12-01819-f006:**
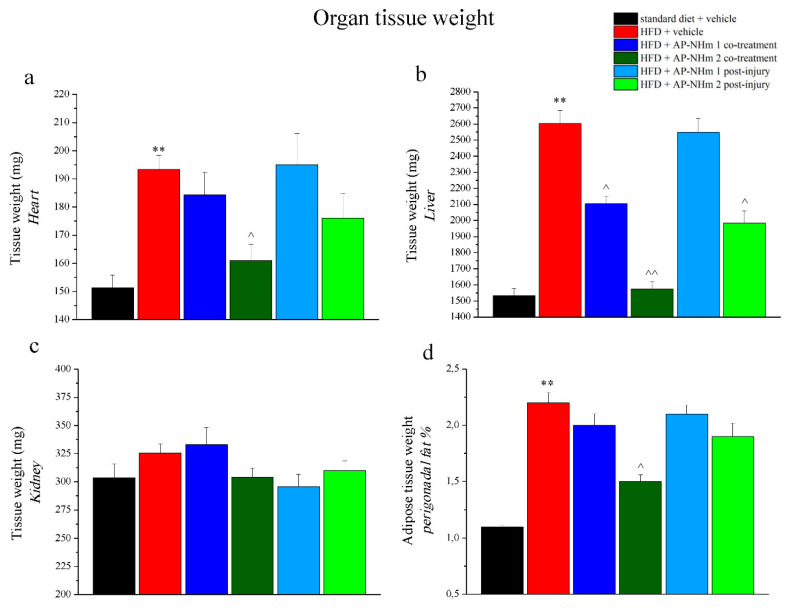
Effects of AP-NHm 1–2, in co-treatment or in post-injury treatment, on organ tissue weight: (**a**) heart, (**b**) liver, (**c**) kidney, (**d**) perigonadal fat pad: on week 12, following in vivo examinations, organs were explanted and weighed. Each value represents the mean ± SEM of 12 mice per group. ** *p* < 0.01 vs. standard diet + vehicle, ^ *p* < 0.05 and ^^ *p* < 0.01 vs. HFD + vehicle.

**Figure 7 nutrients-12-01819-f007:**
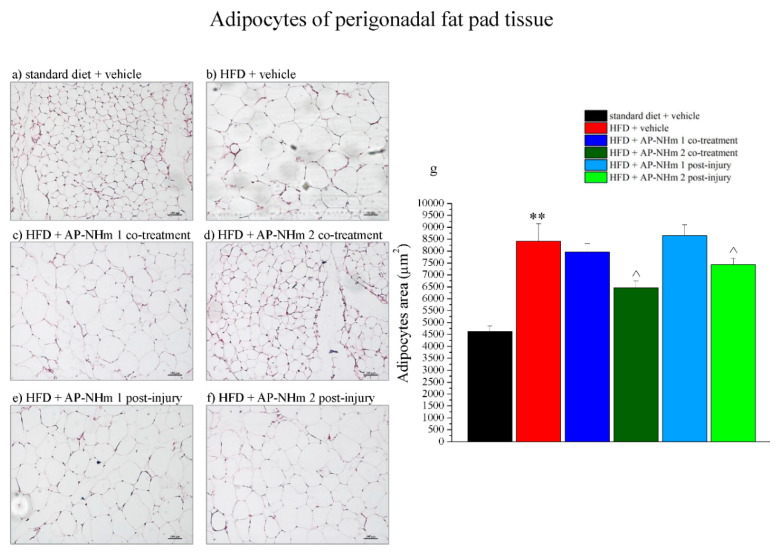
Effects of AP-NHm 1–2, in co-treatment or in post-injury treatment, on adipose tissue and adipocyte area of perigonadal fat pad. (**a**–**f**) Histology of tissue sections of adipose tissue: (**a**) standard diet + vehicle, (**b**) HFD + vehicle, (**c**) HFD + AP-NHm 1 co-treatment, (**d**) HFD + AP-NHm 2 co-treatment, (**e**) HFD + AP-NHm 1 post-injury, (**f**) HFD + AP-NHm 2 post-injury; (**g**) area of adipocytes (µm^2^) in each group of treatment. Each value represents the mean ± SEM of 12 mice per group. ** *p* < 0.01 vs. standard diet + vehicle, ^ *p* < 0.05 vs. HFD + vehicle.

**Figure 8 nutrients-12-01819-f008:**
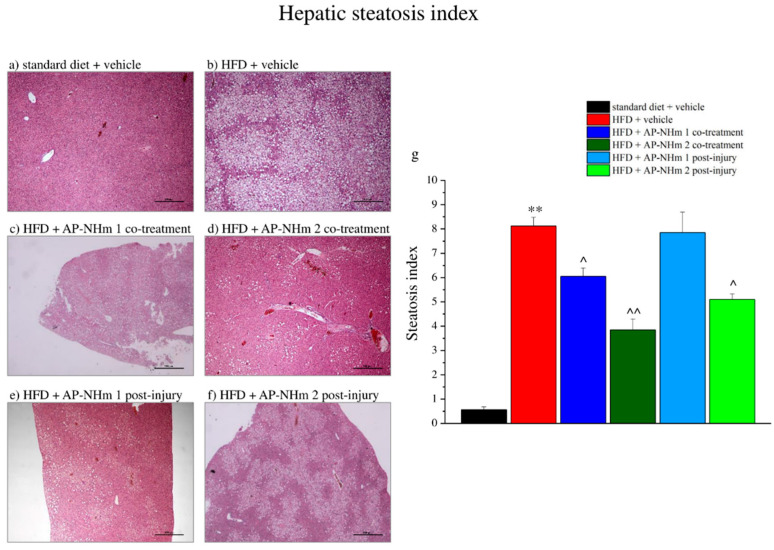
Effects of AP-NHm 1–2, in co-treatment or in post-injury treatment, on hepatic tissue and steatosis index. (**a**–**f**) Histology of tissue sections of hepatic tissue: (**a**) standard diet + vehicle, (**b**) HFD + vehicle, (**c**) HFD + AP-NHm 1 co-treatment, (**d**) HFD + AP-NHm 2 co-treatment, (**e**) HFD + AP-NHm 1 post-injury, (**f**) HFD + AP-NHm 2 post-injury; (**g**) steatosis index of each group of treatment. Each value represents the mean ± SEM of 12 mice per group. ** *p* < 0.01 vs. standard diet + vehicle, ^ *p* < 0.05 and ^^ *p* < 0.01 vs. HFD + vehicle.

**Figure 9 nutrients-12-01819-f009:**
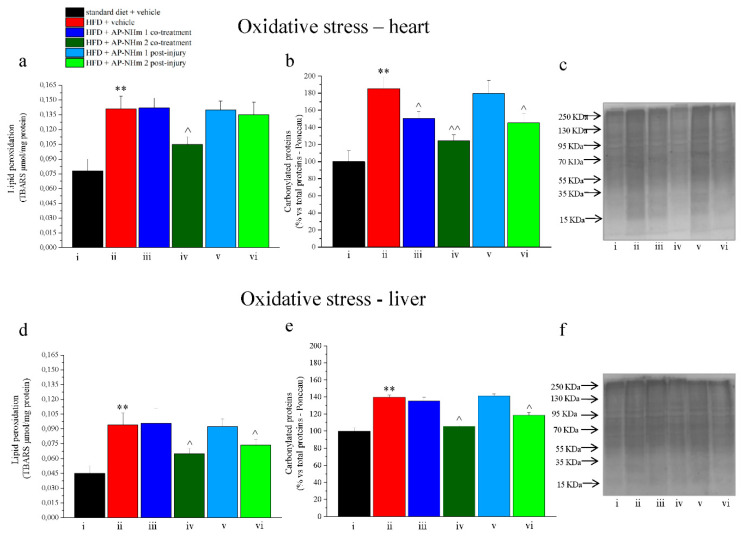
Effects of AP-NHm 1–2, in co-treatment or in post-injury treatment, on oxidative damage in heart (**a**–**c**) and liver (**d**–**f**). (**a**,**d**) Lipid peroxidation measured as TBARS per mg protein; (**b**,**e**) densitometric analysis of carbonylated proteins in total proteins present on Ponceau: results are expressed as percentage of control group (100%); (**c**,**f**) Ponceau on blotting of carbonylated proteins: (**i**) standard diet + vehicle, (**ii**) HFD + vehicle, (**iii**) HFD + AP-NHm 1 co-treatment, (**iv**) HFD + AP-NHm 2 co-treatment, (**v**) HFD + AP-NHm 1 post-injury, **(vi)** HFD + AP-NHm 2 post-injury. Each value represents the mean ± SEM of 12 mice per group. ** *p* < 0.01 vs. standard diet + vehicle, ^ *p* < 0.05 and ^^ *p* < 0.01 vs. HFD + vehicle.

**Table 1 nutrients-12-01819-t001:** Concentration of components in multicomponent food supplement mixtures (AP-NHm): each preparation for in vitro and in vivo experiments.

**In Vitro Experiments**			
**Component**	**AP-NHm 1 (µg/mL)**	**AP-NHm 2 (µg/mL)**	**AP-NHm 3 (µg/mL)**
DHA	1	3	10
Choline Bitartrate	1	3	10
*Tanacetum parthenium* Extract	5	15	50
*Silybum marianum* Extract	5	15	50
Green Coffee Beans Extract	5	15	50
dl α-tocopherol Acetate	0.75	2.5	7.5
**In Vivo Experiments**			
**Component**	**AP-NHm 1 (µg/mL)**	**AP-NHm 2 (µg/mL)**	**AP-NHm 3 (µg/mL)**
DHA	1	3	10
Choline Bitartrate	1	3	10
*Tanacetum parthenium* Extract	5	15	50
*Silybum marianum* Extract	5	15	50
Green Coffee Beans Extract	5	15	50
dl α-tocopherol Acetate	0.75	2.5	7.5

DHA: Docosahexaenoic acid.

**Table 2 nutrients-12-01819-t002:** Effects of OA and AP-NHm mixture on TNF-α and IL-8 concentration (pg/mL) in the cellular medium.

	(pg/mL)	BSA + Vehicle	OA/BSA	AP-NHm 1	AP-NHm 2	AP-NHm 3
Co-treatment	TNF-α	24.3 ± 2.4	94.8 ± 7.4 **	85.1 ± 5.6	60.4 ± 4.9 ^^	54.9 ± 7.2 ^^
	IL-8	12.6 ± 0.9	144.4 ± 10.3 **	112.3 ± 7.8 ^	80.6 ± 6.4 ^^	62.1 ± 8.6 ^^
Post-injury	TNF-α	20.5 ± 1.6	65.3 ± 3.6 **	68.3 ± 5.5	48.1 ± 3.6 ^	40.9 ± 2.2 ^^
	IL-8	18.3 ± 2.0	96.3 ± 8.4 **	96.1 ± 3.4	50.4 ± 4.2 ^^	49.3 ± 5.6 ^

OA: oleic acid; BSA: bovine serum albumin; ** *p* < 0.01 vs. BSA + vehicle, ^ *p* < 0.05 vs. OA/BSA + vehicle, ^^ *p* < 0.01 vs. OA/BSA + vehicle.

**Table 3 nutrients-12-01819-t003:** Effects of OA and AP-NHm mixture on glucose uptake following stimulation with 1-µM insulin.

		BSA + Vehicle	OA/BSA	AP-NHm 1	AP-NHm 2	AP-NHm 3
Co-treatment	Control	6.4 ± 0.6	5.6 ± 0.6	9.8 ± 2.1	5.4 ± 0.6	6.4 ± 3.6
	Insulin 1 μM	128.6 ± 4.3	76.2 ± 4.8 **	97.6 ± 6.7 ^	107.4 ± 6.8 ^^	110.9 ± 8.7 ^^
Post-injury	Control	4.6 ± 1.2	8.8 ± 3.7	3.9 ± 0.9	6.4 ± 2.1	5.4 ± 2.3
	Insulin 1 μM	139.2 ± 10.3	42.6 ± 5.9 **	40.6 ± 14.6	79.1 ± 7.4 ^^	109.8 ± 12.3 ^^

** *p* < 0.01 vs. BSA + vehicle, ^ *p* < 0.05 vs. OA/BSA + vehicle, ^^ *p* < 0.01 vs. OA/BSA + vehicle.

**Table 4 nutrients-12-01819-t004:** Hematic parameters throughout the study: values of cholesterol, triglyceride and glucose from Day 47–Day 84; glucose tolerance test on Day 84. Group 1: standard diet + vehicle; group 2: HFD + vehicle; group 3: HFD + AP-NHm 1 co-treatment; group 4: HFD + AP-NHm 2 co-treatment; group 5: HFD + AP-NHm 1 post-injury; group 6: HFD + AP-NHm 2 post-injury.

**Hematic total cholesterol (mg/dL)**										**Hematic HDL (mg/dL)**		**Hematic LDL** **(mg/dL)**
**Group**	**Day 49**	**Day 56**	**Day 63**		**Day 70**	**Day 77**		**Day 84**		**Day 84**	**Day 84**	
1	102.6 ± 5.1	105.0 ± 5.0	107.0 ± 1.2		105.3 ± 0.9	102.6 ± 2.1		103.8 ± 4.1		51.8 ± 8.5		43.8 ± 6.7
2	140.7 ± 3.4 **	145.8 ± 3.8 **	149.9 ± 2.6 **		155.3 ± 4.8 **	153.8 ± 5.3 **		160.6 ± 3.5 **		60.3 ± 7.0		73.8 ± 5.2 **
3	135.6 ± 4.9	142.8 ± 5.1	145.3 ± 3.7		144.2 ± 4.6 ^	143.1 ± 3.1 ^		150.6 ± 6.8		62.2 ± 5.2		70.6 ± 3.3
4	103.3 ± 3.2 ^^	104.0 ± 4.3 ^^	113.0 ± 5.4 ^^		120.0 ± 6.2 ^^	125.0 ± 4.8 ^^		129.1 ± 4.6 ^^		73.4 ± 2.3 ^		36.9 ± 6.3 ^^
5	142.5 ± 6.2	146.4 ± 6.2	147.3 ± 6.5		145.1 ± 8,3	150.3 ± 4.1		156.9 ± 8.6		59.1 ± 10.5		46.0 ± 9.4 ^^
6	141.9 ± 6.1	144.9 ± 5.3	143.8 ± 3.3		144.1 ± 4.1 ^	142.8 ± 3.2 ^		140.1 ± 5.4 ^^		62.9 ± 4.9		53.2 ± 4.0 ^^
**Triglyceride hematic levels (mg/dL)**												
**Group**	**Day 49**	**Day 56**		**Day 63**			**Day 70**		**Day 77**		**Day 84**	
1	69.0 ± 6.9	71.6 ± 4.4		72.3 ± 6.4			70.5 ± 3.8		69.9 ± 6.1		72.9 ± 5.2	
2	142.3 ± 4.3**	145.7 ± 6.2 **		146.3 ± 4.8 **			151.2 ± 2.2 **		154.4 ± 5.0 **		159.3 ± 7.7 **	
3	115.6 ± 4.3^	114.1 ± 6.6 ^		109.0 ± 5.7 ^^			115.5 ± 3.1 ^		105.6 ± 4.3 ^		100 ± 5.8 ^^	
4	85.8 ± 5.0^	84.2 ± 6.4 ^		86.5 ± 10.7 ^			90.4 ± 7.8 ^^		89.5 ± 5.1 ^^		92.7 ± 5.3 ^^	
5	145.4 ± 7.6	147.1 ± 9.5		143.7 ± 6.7			140.8 ± 6.1		141.2 ± 3.6 ^		139.8 ± 4.4 ^^	
6	142.0 ± 9.0	145.7 ± 5.5		140.9 ± 6.4			130.1 ± 3.2 ^		128.6 ± 5.0 ^^		129.2 ± 3.0 ^^	
**Hematic glucose (mg/dL)**												
**Group**	**Day 49**	**Day 56**		**Day 63**			**Day 70**		**Day 77**		**Day 84**	
1	103.0 ± 2.9	105.7 ± 4.4		101.3 ± 10.1			103.3 ± 4.4		99.6 ± 6.1		105.3 ± 4.6	
2	160.5 ± 6.9 **	155.4 ± 4.2 **		158.3 ± 3.9 **			162.5 ± 6.1 **		164.4 ± 5.1 **		168.3 ± 9.4 **	
3	160.6 ± 8.0	158.3 ± 6.1		163.5 ± 10.7			161.5 ± 5.1		155.0 ± 2.8		166.7 ± 6.5	
4	123.0 ± 5.1^	120.4 ± 5.5 ^		123.9 ± 2.0 ^			133.8 ± 6.0 ^		139.4 ± 6.1 ^		145.9 ± 6.4 ^	
5	152.0 ± 9.9	158.3 ± 6.5		163.3 ± 5.4			159.6 ± 5.9		161.2 ± 8.8		156.3 ± 5.7	
6	153.0 ± 6.2	158.5 ± 8.1		154.8 ± 7.1			146.3 ± 5.5 ^		144.4 ± 3.9 ^		143.8 ± 4.0 ^	
**Glucose tolerance test (Day 84): hematic glucose (mg/mL)**												
**Group**	**Pretest**			**15 min**					**30 min**			
1	105.3 ± 4.6			161.7 ± 3.2					189.0 ± 9.0			
2	168.3 ± 9.4 **			267.0 ± 6.7 **					305.0 ± 5.9 **			
3	166.7 ± 6.5			264.3 ± 8.5					291.2 ± 6.2			
4	145.9 ± 6.4 ^			182.4 ± 9.0 ^					221.8 ± 11.4 ^			
5	156.3 ± 5.7			263.7 ± 10.2					294.7 ± 9.9			
6	143.8 ± 4.0 ^			220.8 ± 10.1 ^					259.1 ± 11.6 ^			

Each value represents the mean ± SEM of 12 mice per group. ** *p* < 0.01 vs standard diet + vehicle, ^ *p* < 0.05 vs HFD + vehicle, ^^ *p* < 0.01 vs HFD + vehicle. HFD: high-fat diet.

**Table 5 nutrients-12-01819-t005:** Effects of HFD and AP-NHm mixture on TNF-α and IL-8 blood concentration on Day 84. Group 1: standard diet + vehicle; group 2: HFD + vehicle; group 3: HFD + AP-NHm 1 co-treatment; group 4: HFD + AP-NHm 2 co-treatment; group 5: HFD + AP-NHm 1 post-injury; group 6: HFD + AP-NHm 2 post-injury.

Group	TNF-α (pg/mL)	IL-8 (pg/mL)
	Day 84	
1	86.4 ± 6.7	249.6 ± 15.9
2	214.8 ± 12.6 **	1087.6 ± 36.3 **
3	164.6 ± 9.5 ^	785.6 ± 40.5 ^
4	120.5 ± 7.5 ^^	405.9 ± 26.8 ^^
5	223.6 ± 16.8	987.9 ± 45.9
6	176.4 ± 6.1 ^	588.7 ± 28.5 ^^

Each value represents the mean ± SEM of 12 mice per group. ** *p* < 0.01 vs standard diet + vehicle, ^ *p* < 0.05 vs. HFD + vehicle, ^^ *p* < 0.01 vs. HFD + vehicle.

**Table 6 nutrients-12-01819-t006:** Effects of HFD and AP-NHm mixture on cholesterol and triglyceride concentrations in liver homogenate on Day 84.

Liver Homogenate Levels Dosages (Day 84)			
Group	LDL (mg/dL)	HDL (mg/dL)	Triglyceride (mg/dL)
1	5.10 ± 0.63	5.42 ± 0.51	93.12 ± 30.34
2	26.16 ± 3.12 **	29.20 ± 3.61 **	602.46 ± 89.32 **
3	21.40 ± 2.34	19.23 ± 1.32 ^	346.67 ± 39.12 ^
4	13.08 ± 1.21 ^^	14.56 ± 2.12 ^	251.70 ± 29.54 ^^
5	27.10 ± 3.29	28.56 ± 4.57	571.34 ± 101.56
6	17.78 ± 1.01 ^	20.03 ± 0.56 ^	359.12 ± 41.34 ^

Each value represents the mean ± SEM of 12 mice per group. ** *p* < 0.01 vs standard diet + vehicle, ^ *p* < 0.05 vs. HFD + vehicle, ^^ *p* < 0.01 vs. HFD + vehicle.

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
