# Peer review of "Treatment of Non-Alcoholic Steatosis: Preclinical Study of a New Nutraceutical Multitarget Formulation"

_nutrients, 2020, doi:10.3390/nu12061819_

Round 1

Reviewer 1 Report

In the manuscript submitted, Authors described a study efficacy, in both an in vitro and in vivo experimental model of NASH, using a nutraceutical supplement diet formulation based on various sources, mainly from vegetables, whose anti-inflammatory properties where singularly and partially investigated in previous studies.

This paper meets with the aims of the journal and investigated with accuracy the benefical effects of this new compound in the prevention and therapy of NASH, a multi-factorial metabolic syndrome that is growing with an alarming rate at world level.

The topic of the research is interesting and fit completely with other trending papers about the subject; the paper considers and describes adequately the “state of the art” on the matter about the use of the formulation and its potential and benefical effects in the therapy of NASH.

The Title and the Abstract appear to to be clearly formulated and are absolutelly consisted with the rest of the manuscript.

Regard the title, I would suggest to explicit that the aim of the study is about a nutraceutical formulation since then, in the proposed sentence it may seem too generic,

The Experimental approaches are consistent with the hypothesis and in particular, the in vivo model of NASH appear to be deeply and rigorously managed and investigated and  the results clearly stated in a manner consistent with the experimental approach and research objectives.

In the Discussion chapter, Authors have critically discussed their results adequately and coherently with the aim of the study as expressed in introduction and the frame of experimental design.

Illustrations, graphs and tables are clear and correct and references are faithfully cited

I consider this paper intriguing , rigorous and outstanding and I will suggest to the editors to publish without any changes except for a different formulation of the titlein order to focusung better with the aim of the journal.

Author Response

Response to Reviewer 1 Comments

In the manuscript submitted, Authors described a study efficacy, in both an in vitro and in vivo experimental model of NASH, using a nutraceutical supplement diet formulation based on various sources, mainly from vegetables, whose anti-inflammatory properties where singularly and partially investigated in previous studies.

This paper meets with the aims of the journal and investigated with accuracy the benefical effects of this new compound in the prevention and therapy of NASH, a multi-factorial metabolic syndrome that is growing with an alarming rate at world level.

The topic of the research is interesting and fit completely with other trending papers about the subject; the paper considers and describes adequately the “state of the art” on the matter about the use of the formulation and its potential and benefical effects in the therapy of NASH.

The Title and the Abstract appear to to be clearly formulated and are absolutelly consisted with the rest of the manuscript.

Regard the title, I would suggest to explicit that the aim of the study is about a nutraceutical formulation since then, in the proposed sentence it may seem too generic,

The Experimental approaches are consistent with the hypothesis and in particular, the in vivo model of NASH appear to be deeply and rigorously managed and investigated and  the results clearly stated in a manner consistent with the experimental approach and research objectives.

In the Discussion chapter, Authors have critically discussed their results adequately and coherently with the aim of the study as expressed in introduction and the frame of experimental design.

Illustrations, graphs and tables are clear and correct and references are faithfully cited

I consider this paper intriguing , rigorous and outstanding and I will suggest to the editors to publish without any changes except for a different formulation of the titlein order to focusung better with the aim of the journal.

Response 1:

Dear Reviewer 1,

thank you for your kind comments on our paper and for your suggestion on the title. We accepted your suggestion and we therefore change the title to “Treatment of non-alcoholic steatosis: preclinical study of a new nutraceutical multitarget formulation”.

Reviewer 2 Report

This is an in vitro and in vivo study on the effects on lipid metabolism, oxidative stress parameters, and fatty liver of AP-NH which is a combination of anti-oxidants, omega 3 fatty acids, and green coffee extract. Both the in vitro (on hep G2 cells) and the in vivo experiments (mice) were conducted with a study design which included a prevention subgroup and a treatment subgroup (provided after the development of cellular or tissue damage). The study showed that AP-NH effects varied according to the dose and the compound was mostly effective in the prevention arm, but beneficial effects were observed also in the post-treatment. The study is informative and provide data on a new possible treatment for nonalcoholic liver disease. Fatty liver and hepatic steatosis improved after the proposed intervention. The use of a new multitarget formulation may address several aspects or mechanisms of damage in fatty liver. However,. it also prevent the determination of which components among the various compounds is producing the major effect.

The manuscript should be improved with the following changes:

1) the authors should provide liver/body weight ratio

2) the introduction and the discussion in some parts provide redundant information which should be shortened

3) some information are lacking. For example, data on plasma transaminase levels, quantification of inflammation in the liver histology

Author Response

Reviewer 2

  • Point 1: the authors should provide liver/body weight ratio.

We calculated the liver/body ratio as reported in the table below, the data did not seem informative so, if the referee agrees, we preferred not to add this value to the text.

Liver/body weight ratio %

liver w.

body w.

ratio

Ctrl

15

32,6

4,699387

HFD

2,6

46,5

5,599570

prev LOW

2,1

44

4,780455

prev HIGH

1,6

41,8

3,767943

ther LOW

2,5

46

5,536957

ther HIGH

2,0

44

4,511364

  • Point 2: the introduction and the discussion in some parts provide redundant information which should be shortened.

We cut on some parts in the introduction and the discussion which may seem redundant at lines 57-61, 558-560, 650-651.

  • Point 3: some information are lacking. For example, data on plasma transaminase levels, quantification of inflammation in the liver histology.

Unfortunately, we haven’t got these data, therefore we cannot provide them since we have not available tissues. The realization of new in vivo experiments to obtain tissues is very difficult in this moment. We will take in consideration these parameters to be investigated in the future research.

Reviewer 3 Report

Overall, the manuscript was badly prepared, with lots of grammar errors, low quality of figures.

  1. Abstract needs some background introduction for the multicomponent food supplement mixture, such as its components and targets, and what does ratio means. Before reading the maintext, this reviewer could not understand what treatment the authors did.
  2. In introduction, regional and sex-related differences part is not very necessary.
  3. For the in vitro model of NASH, oleic acid treatment is not an idea model. In the three literature cited, oleic acid were used to induce lipogenesis and steatosis, instead of hepatocyte or liver injury.
  4. For in vivo treatment, 7 weeks of 60% HFD treatment is not very likely to induce NASH.
  5. Please move the introduction about in vitro and in vivo treatment into “Materials and Methods”. What’s the ratio of each component in the AP-NHm?
  6. In the “Results”, title each part with your observations or conclusions, instead of “In vitro experiments”.
  7. Figure legends need more information, about your treatment details. Some readers go through your paper by reading your figures. Make sure they can understand what you did and your resuts.
  8. In figure 2, why the panel label “b” and “d” are on the top-right of its panel?
  9. The rationale that those components are included should be specified. And to verify whether all the components of this mixture contribute significantly to the improvement of NASH, it will be interesting to test the effect of other combinations. For example, if one or two of the components are removed, does the mixture still work?

Author Response

Reviewer 3

  • Point 1: Abstract needs some background introduction for the multicomponent food supplement mixture, such as its components and targets, and what does ratio means. Before reading the maintext, this reviewer could not understand what treatment the authors did.

The main effects of the compounds we used were anti-dislipidaemic, antioxidant and anti-inflammatory. We cannot anticipate the list of the compounds in the abstract because they are too many to be written there without exceeding the word limit, but we summarized their main effects at line 20.

  • Point 2: In introduction, regional and sex-related differences part is not very necessary.

Following it, we reduced the part about regional and sex-related differences (lines 43-45).

  • Point 3: For the in vitro model of NASH, oleic acid treatment is not an idea model. In the three literature cited, oleic acid were used to induce lipogenesis and steatosis, instead of hepatocyte or liver injury.

We see the point of the reviewer point, but lipogenesis and steatosis are actually hallmarks of steatohepatitis and steps of the multiple-hit pathogenesis of NASH. The cited paper of Vidyashankar S. and his collaborators (Toxicol Vitr. 2013, 27, 945–953) supports this model along with many others in literature, such as Ziamajidi N. et al. (Food Chem Toxicol, 2013, 58, 198-209). We were also well-aware of the numerous limitations which an in vitro model implicates, and those limitations were overcome by testing the nutraceutical mixture also in an in vivo model of NASH.

  • Point 4: For in vivo treatment, 7 weeks of 60% HFD treatment is not very likely to induce NASH.

We agree that the period of treatment is quite short to induce NASH. Nevertheless, probably by the murine fast metabolism, this kind of treatment allowed us to obtain alterations reported in the present study (both in results and in supplementary text), which clearly show alteration of lipid and carbohydrate metabolism together with histological hepatic damage. This data agree with other already published papers cited in the Methods.

  • Point 5: Please move the introduction about in vitro and in vivo treatment into “Materials and Methods”. What’s the ratio of each component in the AP-NHm?

Following this suggestion, we moved the part of introduction about in vitro and in vivo treatment into “Materials and Methods” (lines 86-99; 155-163). The ratio of each component in the AP-NHm, that was displayed in supplementary files, was moved to the “Materials and Methods” as well (table 1 and table 2).

  • Point 6: In the “Results”, title each part with your observations or conclusions, instead of “In vitro experiments”.

Following this suggestion, we titled each part of “Results” (lines 303, 314, 334, 345, 356, 376, 389, 414, 449, 461, 485, 497, 520). 

  • Point 7: Figure legends need more information, about your treatment details. Some readers go through your paper by reading your figures. Make sure they can understand what you did and your resuts.

Following this suggestion, we add more details in figure legends (lines 310-312; 329-332; 371-375; 383-385;408-412; 447; 456-458; 459-460; 482-484; 495-496; 509-510; 518; 533-534; 537).

  • Point 8: In figure 2, why the panel label “b” and “d” are on the top-right of its panel?

The position of the panel label “b” and “d” is on the top-right of its panel also in figure 3. The labels were put there because they seemed otherwise too near to the title of the graph. Nevertheless, we followed the suggestion of this reviewer and we moved them to the top-left of its panels.

  • Point 9: The rationale that those components are included should be specified. And to verify whether all the components of this mixture contribute significantly to the improvement of NASH, it will be interesting to test the effect of other combinations. For example, if one or two of the components are removed, does the mixture still work?

The rationale behind the inclusion of each component is explained in the abstract, the introduction and in the methods. We decided to investigate the beneficial effects of a multicomponent supplementary mixture because of the multi-etiological profile of NASH. Moreover, it seemed to us not necessary moving on further evaluations about fractioned mixtures, in compliance with the principle of 3R (Replacement, Reduction and Refinement).

Round 2

Reviewer 3 Report

They revised the manuscript well.